# Characterization of the Bacterial Profile from Natural and Laboratory *Glossina* Populations

**DOI:** 10.3390/insects14110840

**Published:** 2023-10-29

**Authors:** Youssef El Yamlahi, Naima Bel Mokhtar, Amal Maurady, Mohammed R. Britel, Costas Batargias, Delphina E. Mutembei, Hamisi S. Nyingilili, Deusdedit J. Malulu, Imna I. Malele, Elias Asimakis, Panagiota Stathopoulou, George Tsiamis

**Affiliations:** 1Laboratory of Innovative Technologies, National School of Applied Sciences of Tangier, Abdelmalek Essaâdi University, Tétouan 93000, Morocco; elyamlahi.youssef@gmail.com (Y.E.Y.); naima.belmokhtar@upatras.gr (N.B.M.); amal.maurady.ma@gmail.com (A.M.); mbritel@uae.ac.ma (M.R.B.); 2Faculty of Sciences and Technics of Tangier, Abdelmalek Essaâdi University, Tétouan 93000, Morocco; 3Laboratory of Systems Microbiology and Applied Genomics, Department of Sustainable Agriculture, University of Patras, 2 Seferi St, 30131 Agrinio, Greece; eliasasim@gmail.com (E.A.); panayotastathopoulou@gmail.com (P.S.); 4Department of Biology, University of Patras, 26504 Patras, Greece; cbatargias@upatras.gr; 5Vector & Vector Borne Diseases, Tanzania Veterinary Laboratory Agency (TVLA), Tanga P.O. Box 1026, Tanzania; phinadel5@gmail.com (D.E.M.); mnyingilili@gmail.com (H.S.N.); maluluone@gmail.com (D.J.M.); 6Directorate of Research and Technology Development, TVLA, Dar Es Salaam P.O. Box 9254, Tanzania; maleleimna@gmail.com

**Keywords:** tsetse fly, bacterial profile, mass rearing, 16S rRNA, amplicon sequencing

## Abstract

**Simple Summary:**

Tsetse flies are large biting insects that inhabit much of tropical Africa and have a significant economic impact as the biological vectors of trypanosomes, which cause serious diseases to humans and livestock. A large array of bacteria, termed collectively symbionts, inhabit the internal organs of the flies’ body. These bacterial symbionts are involved in important aspects of the flies’ biology, including nutrition and reproduction. For instance, the main bacterial symbiont *Wigglesworthia* provides nutritional supplements necessary for host fertility and development, while *Wolbachia* is known to affect the reproduction of flies by causing a series of abnormalities. Therefore, the symbionts of tsetse flies show promising signs for exploitation and can be used for the development of innovative tools for the control of the flies and the diseases they carry. In this work, we used next-generation sequencing to characterize in detail the bacterial communities of four tsetse fly species. Their bacterial communities differed significantly, depending on the origin and the developmental stage of the flies. Certain important bacteria, such as *Wigglesworthia* and *Sodalis*, were present in all species and exhibited a high number of interactions with the other members of the bacterial community. Finally, *Wolbachia* was mostly present in *G. morsitans* samples.

**Abstract:**

Tsetse flies (*Glossina* spp.; Diptera: Glossinidae) are viviparous flies that feed on blood and are found exclusively in sub-Saharan Africa. They are the only cyclic vectors of African trypanosomes, responsible for human African trypanosomiasis (HAT) and animal African trypanosomiasis (AAT). In this study, we employed high throughput sequencing of the 16S rRNA gene to unravel the diversity of symbiotic bacteria in five wild and three laboratory populations of tsetse species (*Glossina pallidipes*, *G. morsitans*, *G. swynnertoni*, and *G. austeni*). The aim was to assess the dynamics of bacterial diversity both within each laboratory and wild population in relation to the developmental stage, insect age, gender, and location. Our results indicated that the bacterial communities associated with the four studied *Glossina* species were significantly influenced by their region of origin, with wild samples being more diverse compared to the laboratory samples. We also observed that the larval microbiota was significantly different than the adults. Furthermore, the sex and the species did not significantly influence the formation of the bacterial profile of the laboratory colonies once these populations were kept under the same rearing conditions. In addition, *Wigglesworthia*, *Acinetobacter*, and *Sodalis* were the most abundant bacterial genera in all the samples, while *Wolbachia* was significantly abundant in *G. morsitans* compared to the other studied species. The operational taxonomic unit (OTU) co-occurrence network for each location (VVBD insectary, Doma, Makao, and Msubugwe) indicated a high variability between *G. pallidipes* and the other species in terms of the number of mutual exclusion and copresence interactions. In particular, some bacterial genera, like *Wigglesworthia* and *Sodalis*, with high relative abundance, were also characterized by a high degree of interactions.

## 1. Introduction

Tsetse flies are large biting insects that inhabit much of tropical Africa [1,2]. Regarding their taxonomical distribution, tsetse species are included within the genus *Glossina* and the family Glossinidae. They are obligate parasites that live by feeding on the blood of vertebrate animals. Due to this distinct feeding mechanism, tsetse flies have been considerably studied because of their unique role in transmitting the vector-borne pathogen. They have a prominent economic impact in sub-Saharan Africa as the biological vectors of infectious species of *Trypanosoma* that are responsible for causing severe diseases such as sleeping sickness (human African trypanosomiasis, HAT) and animal African trypanosomiasis (AAT) [1,2]. Several thousand HAT cases are recorded annually, and recent conservative estimates indicate that more than 50 million people are at risk [3]. Additionally, it is predicted that AAT causes agricultural economic losses of around USD 5 billion per year [4]. Fexinidazole, which was initially identified as a broad-spectrum anti-infective agent [5], is regarded as a valuable treatment for early-stage HAT [6,7,8]. It demonstrates similar effectiveness and safety as the best current treatment for HAT, the nifurtimox/eflornithine combination therapy (NECT) [9,10], potentially replacing it and becoming the first orally administered monotherapy for *T*. *brucei gambiense* HAT [11].

Tsetse larvae feed on the milk produced in their mother’s milk glands and become pupae within an hour after birth. Adult flies of both sexes feed largely on blood meals. The tsetse microbiota is intriguing due to the flies’ unique lifestyle, bilateral transmission from both parents to their offspring, involvement in reproductive strategies like cytoplasmic incompatibility, and its potential for vector and disease control [12,13,14].

There is an urgent need to develop alternative and effective control methods that will have a minimal impact on the environment, cover the lack of effective vaccines and medical treatment, and overcome the deficiencies of traditional control strategies that rely heavily on the use of chemical insecticides and mostly inefficient fly traps [15]. The sterile insect technique (SIT), an environmentally friendly technique that is based on mass-rearing and releasing flies that have been previously rendered sterile via ionizing radiation, has proven to be successful in eradicating tsetse flies from Unguja Island (Zanzibar), Tanzania, where it was used in the frame of area-wide integrated pest management programs (AW-IPM) [16]. Sterile male individuals are commonly used during this technique. Notably, the application of radiation has damaging effects on the insects, reducing their fitness and competition against wild individuals [17]. Cultivable gut bacteria with nutritional value or probiotic action, which are part of the natural flora of wild populations, could be supplemented in artificial feed to improve the overall quality of sterile males [18,19].

So far, tsetse symbiosis has been focused around four core bacterial genera: *Wigglesworthia*, *Sodalis*, *Wolbachia*, and *Spiroplasma*. These four symbionts form the so-called dogma of the tsetse symbiosis. The main mutualistic symbiont *Wigglesworthia* supports host fertility, larval development, and the maturation process of the adult immune system by providing necessary nutrients to the host [15,20,21,22]. The presence of the facultative symbiont *Sodalis* in tsetse populations is putatively related to the vectorial capacity of the fly and, more specifically, its ability to transmit trypanosomes [23]. Various prevalences of the third symbiont, *Wolbachia*, have been seen in tsetse fly natural populations. Certain species exhibited infection rates of up to 100% [24,25], while others were found to harbor no *Wolbachia* infections at all, such as *G. palpalis palpalis* (Gpp) [25]. Additionally, the *Wolbachia* strain present in *G. morsitans morsitans* (Gmm) can disrupt the fly’s normal reproductive activity by inducing cytoplasmic incompatibility [24], a reproductive phenomenon in some insects involving incompatibility in symbiotic microorganisms, often leading to reduced fertility or increased offspring mortality when individuals with different microbial strains mate [26]. The fourth symbiont is *Spiroplasma*, representing a new class of tsetse symbionts found in *G. fuscipes fuscipes* (Gff), *G. tachinoides*, and Gpp, which is also related to the induction of reproductive phenotypes to its host [27]. *Spiroplasma* is allocated mostly in Gff testes in higher concentrations compared to female flies’ ovaries. Interestingly, except for its role as a reproductive parasite, a putative mutualistic role was acknowledged for certain *Spiroplasma* strains [27].

Undoubtedly, the microbiota of tsetse flies shows promising signs for exploitation since it is involved in several aspects of host biology and can be used for the development of innovative tools and strategies for vector and disease control [14,15,20]. So far, the microbial composition of both laboratory and natural populations of different species of tsetse flies has been characterized by either deploying culture-dependent or molecular-based methods [27,28,29]. These studies have managed to identify several bacterial genera, including *Enterobacter*, *Enterococcus*, and *Acinetobacter* strains, in various tsetse species either while studying isolated tissues (e.g., the gastrointestinal tract or reproductive tissues) or whole flies [30,31]. Notably, culture-dependent approaches resulted in the isolation of a new bacterial species, *Serratia glossinae*, from the midgut of *Glossina palpalis gambiensis* (Gpg) insectary flies [32]. The standalone use of molecular techniques and next-generation sequencing, or their combined use with culture-dependent techniques, improved the depth of the analysis of tsetse bacterial communities [29,33]. Recently, our group used 16S rRNA amplicon sequencing to identify for the first-time multiple occurrences of *Spiroplasma* infections within samples originating from wild populations of Gff, Gpp, and *G. tachinoides* (Gt), which are all members of the palpalis subgroup of the *Glossina* genus [27]. Additionally, the same study identified members of the Enterobacteriaceae family, such as *Klebsiella*, *Erwinia*, and *Serratia*, in the *Glossina* species from Burkina Faso, including *G. medicorum* (Gmed), *G. morsitans submorsitans* (Gms), Gpg, and Gt. Apart from wild populations, the analysis also involved the bacterial communities of laboratory-reared Gff, *G. morsitans*, and *G. pallidipes* [27].

The focus of this current study is the analysis of the symbiotic bacterial profiles of wild and laboratory populations of different tsetse species. In total, this study is centered around five wild and three laboratory populations of four tsetse species, *G. pallidipes*, *G. morsitans*, *G. swynnertoni*, and *G. austeni*. The bacterial profiles were investigated using next-generation amplicon sequencing of the V3-V4 region of the 16S rRNA gene. The end goal was to assess the dynamics of the bacterial community within each laboratory and wild population in relation to various parameters of the flies, including the developmental stage, their age, sex, and location of origin. The comparative analysis between wild and lab samples could potentially reveal interesting bacterial players that could be tested for putative probiotic action. Promising candidates could be introduced to the mass-rearing process to improve it. Moreover, the type and number of interactions that develop between bacterial partners within the symbiotic community were characterized. Networks could reveal bacterial taxa that prefer to co-exist and could display synergistic functionality within a host or taxa that prefer to disassociate.

## 2. Materials and Methods

### 2.1. Sample Overview

The samples studied were classified according to species, location, sex, and developmental stage. We analyzed 159 samples from four *Glossina* species, *G. pallidipes*, *G. morsitans morsitans*, *G. swynnertoni*, and *G. austeni*, collected from the wild (Msubugwe forest reserve in Pangani district, Makao in Serengeti National Park, and Doma from Morogoro) in Tanzania and from laboratory populations that were kept at the Vector and Vector-Borne Diseases Insectary (VVBD Insectary). All samples from the wild were adult male flies (whole body), while the samples from the laboratory populations were dissected, and the gastrointestinal tract was collected from larvae or adults (female and male) (Table 1). Each lab sample was a pool of five tissues, while wild samples consisted of individual insects. Laboratory samples from different developmental stages were selected to reveal differences related to the mass-rearing process (e.g., due to the different feed provided at each stage) or the biology of the insects. Such biological parameters may include the developmental process itself (e.g., creation of new niches for bacteria during the formation of tissues and organs) and the unique requirement for nutrients in each stage. The age of tsetse flies also affects trypanosome infection [34]. For wild populations, the analysis was focused on males mainly due to their role in SIT applications. Similarly, for lab samples, the analysis was focused on the gut communities as they are of specific interest for mass breeding programs.

### 2.2. DNA Extraction and PCR Amplification

The *Glossina* samples were surface sterilized with a 70 percent *v*/*v* ethanol solution and washed with sterile water prior to DNA extraction. A modified protocol based on Cetyl Trimethyl Ammonium Bromide (CTAB) was used to isolate DNA [35]. The quality and quantity of the extracted DNA samples were tested with a Q5000 Microvolume UV-Vis spectrophotometer (Quawell Technology, San Jose, CA, USA). DNA samples were kept at −20 °C until PCR amplification. PCR amplification of the variable V3-V4 region of the 16S rRNA gene was performed in 25 µL reactions using primers U341F-MiSeq 5′-TCGTCGGCAGCGTCAGATGTGTATAAGAGACAG(CCTACGGGRSGCAGCAG)-3′ and U805R-MiSeq 5′-GTCTCGTGGGCTCGGAGATGTGTATAAGAGACA(GGACTACHVGGGTATCTAATCC)-3′ (parentheses indicate the primer region) [36] and the KAPA Taq PCR Kit (Roche, Basel, Switzerland). Each reaction contained 2.5 µL of 10× KAPA Taq buffer, 0.2 µL a mixed solution of dNTPs (25 mM each), 0.1 µL of KAPA Taq DNA Polymerase (5 U/μL), 1 µL of each primer (10 µM), 1 µL of template DNA solution, and 19.2 µL of sterile deionized water. The temperature profile of the PCR included an initial denaturation step at 95 °C for 3 min, followed by 35 cycles of 95 °C for 30 s, 54 °C for 30 s, 1 min at 72 °C, and a final extension cycle at 72 °C for 5 min.

The PCR products were separated via electrophoresis on a 1.5% (*w*/*v*) agarose gel in TAE buffer (40 mM Tris-acetate, 1 mM EDTA). The desired 550 bp amplification product was visualized in Bio-Rad’s Gel Doc XR + system. The polyethylene glycol technique (20% PEG, 2.5 M NaCl) [37] was used to purify positive PCR results before they were resuspended in 15 μL of sterile water. The quality and quantity of purified PCR products were evaluated using a Q5000 Microvolume UV-Vis spectrophotometer as proceeded after DNA extraction (Quawell Technology, San Jose, CA, USA).

### 2.3. Index PCR and Illumina Sequencing

The purified PCR products were diluted to a final concentration of 10 ng/μL and submitted to index PCR to incorporate the Illumina barcodes and adaptors. Each sample contained a unique combination of index primers to assist the demultiplexing process. Samples were then pooled equimolarly and sequenced by Macrogen, Inc. (Seoul, Republic of Korea), using a 2 × 300 bp paired-end kit on a MiSeq platform.

In more detail, the amplification reaction (50 μL) was performed using the KAPA Taq PCR Kit (Roche, Basel, Switzerland). Each reaction contained 5 μL of KAPA Taq Buffer (10×), 0.4 μL of a mixed dNTPs solution (25 mM each), 5 μL of the forward index primer (10 μM), 5 μL of the reverse index primer (10 μΜ), 0.2 μL of KAPA Taq DNA Polymerase (5 U/μL), 2 μL of the diluted PCR product (10 ng/μL), and 32.4 μL of sterile deionized water. The PCR program consisted of an initial denaturation cycle at 95 °C for 5 min, followed by 8 cycles of denaturation at 95 °C for 30 s, annealing at 55 °C for 30 s, extension at 72 °C for 1 min, and a final extension cycle at 72 °C for 5 min. The generated amplicons were purified using the NucleoMag^®^ NGS Clean-up and Size Selection kit, following the manufacturer’s instructions (Macherey-Nagel, Düren, Germany). A Quawell Q5000 microvolume UV-Vis spectrophotometer was used to determine the concentration of purified samples after they had been suspended in 30 μL of sterile deionized water. All samples were diluted to a final concentration of 8 nM, and equimolar volumes were pooled to prepare the library for Illumina MiSeq sequencing. All data generated from this study can be retrieved from the SRA database of NCBI under BioProject accession number PRJNA950333.

### 2.4. Data Analysis

Raw sequencing reads were pre-processed using usearch version 11 [38]. The paired-end reads were assembled to create consensus sequences and trimmed by length using fastq-mergepaired command. Then, the quality of assembled reads was improved using fastq-filter command based on expected error numbers (e.e) (reads were discarded if e.e > 1%). The resulting sequences were clustered into operational taxonomic units (OTUs) at 97% sequence similarity with the cluster-otus command of the UPARSE-out algorithm [39], and singleton unique sequences were discarded. An OTUs table was generated using otutab command. Incorrectly assigned OTUs were identified and filtered using uncross command based on the UNCROSS2 algorithm [40]. The OTU table was trimmed by removing low abundance counts, samples, and OTUs based on OTUs trim threshold parameter (minimum size for an OTU as a fraction of all OTUs). The trimmed OTU sequences were extracted by matching sequence labels and then imported to Qiime2 [41] to be assigned taxonomy based on percentage identity using BLAST+ and the SILVA (16S) database, release 128, as reference [42]. A phylogenetic tree was constructed with FastTree [43]. The “diversity” function of the R package “vegan” [44] was used for alpha diversity analyses. Alpha diversity indices were plotted using “ggplot” function from package “ggplot2” [45]. The non-parametric Kruskal–Wallis Rank Sum Test with the Wilcoxon Rank Sum Test were used to assess statistical differences in bacterial composition and relative abundance between various groups of samples [46]. Beta diversity was calculated using generalized UniFrac [47]. Visualization of the multidimensional distance matrix in two dimensions was performed via nonmetric Multi-Dimensional Scaling (NMDS) [48].

Co-occurrence networks were used for the visualization of interactions between microorganisms. These interactions, which do not often include physical contact, refer to microorganisms that execute related or complementary tasks and/or share environmental niches [49,50]. Co-occurrence networks of OTUs were generated using CoNet, a plugin for Cytoscape 3.8.2. (Institute for system biology, Seattle, WA, USA) [51]. Network graphs were produced using Gephi 0.9.2 (Gephi, WebAtlas, Paris, France). The network was built based on the Pearson and Spearman correlation coefficients, Mutual Information, Bray–Curtis, and Kullback–Leibler dissimilarity indices. Statistical significance of different interactions (copresence or mutual exclusion) was computed with edge-specific permutation and bootstrap score distributions, with 1000 iterations. Edges with initial scores outside the bootstrap distribution’s 0.95 range were eliminated, and *p*-values were adjusted using the Benjamini–Hochberg technique. In each network graph, nodes correspond to microbial OTUs and edges to bacterial associations. The size of each node is proportional to the number of interactions with other nodes.

## 3. Results

### 3.1. Overall Data Analysis

After a strict trimming process, a total of 9,069,871 high-quality paired-end reads were obtained from 159 samples with an average of 57,043 reads/sample. In total, 29 unique OTUs were identified with a relative abundance greater than 0.1% (Table 1). They were classified into three Phyla: Proteobacteria, Firmicutes, and Actinobacteria. Proteobacteria was the most abundant taxonomic group (97.8% of reads), followed by Firmicutes (1.9%) and Actinobacteria (0.3%). Within Proteobacteria, two taxonomic classes dominated: Alpha- and Gammaproteobacteria, while Firmicutes was represented by Bacilli. At the genus level, *Wigglesworthia* and *Sodalis* were the most abundant taxa (70% and 17.4% of the obtained reads, respectively), followed by *Wolbachia* (2.5%), *Acinetobacter* and *Pseudocitrobacter* (both at 2.1%), and *Pseudomonas* (1.3%) (Appendix A).

### 3.2. Laboratory Populations Display a Species-Specific Bacterial Profile

There were significant differences in bacterial composition within each laboratory population (Permanova; *p* < 0.001; Appendix A). In addition to the most common bacterial genera, such as *Wigglesworthia* and *Sodalis*, we found that certain OTUs are more prevalent in some species than others. Such is the case for *Phyllobacterium*, *Pseudomonas* and *Psychrobacter* which are more present in *G. austeni* (3 ± 1.85%, 1.12 ± 0.73% and 1.99 ± 1.24%, respectively) and *G. morsitans morsitans* (3.07 ± 0.97%, 1.36 ± 0.87% and 1.62 ± 0.97%, respectively) than *G. pallidipes* (0.48 ± 0.33%, 0.16 ± 0.11% and 0.22 ± 0.13%, respectively). On the contrary, *Bacillus* and *Lysinibacillus* are more prevalent in *G. pallidipes* (1.51 ± 1.35% and 1.27 ± 1.1%, respectively) than the other species (from 0.02 ± 0.02% to 0.82 ± 0.45%). Some bacteria were only found in one species, although in small relative abundance, such as *Acinetobacter* (0.009%) in *G. austeni* and *Enterococcus* (0.056%) in *G. morsitans morsitans* (Appendix A).

The differences In the bacteriota of *G. austeni* were displayed in the pCoA analysis, where larval samples formed a separate cluster compared to adult samples (1, 5, and 15 days), which formed a tight cluster (Permanova; *p* < 0.002; Figure 1). The Permanova analysis also revealed that the bacterial community was quite similar at 1, 5, and 15 days (Permanova; *p* > 0.05; Appendix A). However, compared to adults, larvae had significantly higher Shannon index values. Statistical differences were also found for richness. However, there are no significant differences in evenness (Pielou index) or Simpson index between larva and 5-day midgut. (Appendix A).

The bacterial profile of *G. austeni* was dominated by Proteobacteria (from 95.57 to 99.99%), followed by Firmicutes mainly in larvae and 5-day-old adults, while a minor Actinobacteria community was identified in larvae (Figure 2A). Gammaproteobacteria was the most abundant class across the developmental stages, followed by Alphaproteobacteria, which was highly abundant only in larvae (Figure 2B). At genus level, *Wigglesworthia* was highly abundant in adult flies (98.9 ± 0.53%, 74.4 ± 10.58%, and 97.4 ± 1.56% in 1, 5, and 15 days, respectively), while in larvae, a high abundance of different bacterial genera were observed, including *Wigglesworthia* (30.3 ± 5.99%), *Phyllobacterium* (26.2 ± 3.34%), and *Psychrobacter* (17.3 ± 2.86%) (Figure 2C).

The *G. morsitans morsitans* laboratory population differences were also displayed in PCoA analysis, where larval samples formed a separate cluster compared to adult samples (Permanova; *p* < 0.001; Figure 3). The Permanova analysis revealed that the bacterial community among different adult stages was similar (Permanova; *p* > 0.05; Appendix A), except for 1- and 15-day-old adults (Permanova; *p* < 0.05; Appendix A). The larvae showed significantly higher values of the Shannon index than other developmental stages. Statistical differences were also observed for the other three alpha diversity indices. For adult flies, significant differences were only observed for the richness index, between 5- and 15-day samples (Appendix A).

*Glossina. morsitans morsitans* laboratory population was also dominated by Proteobacteria in all developmental stages examined (from 96.24% to 99.99%) followed by Firmicutes and Actinobacteria (Figure 4A). Gammaproteobacteria was the most abundant class across the developmental stages, followed by Alphaproteobacteria, which was highly abundant in the larvae (Figure 4B), with the Actinobacteria identified only at the larval stage. At genus level, *Wigglesworthia* was very abundant in the adult flies (93.8 ± 2.01%, 83.2 ± 5.54% and 90.6 ± 1.49% in 1, 5, and 15 days, respectively), while in the larvae, a multitude of different bacterial genera was observed, including *Wiggleswothia* (44.6 ± 10.84%), *Phyllobacterium* (19 ± 7.46%), *Psychrobacter*, *Sodalis*, etc. (Figure 4C).

*Glossina pallidipes* laboratory population differences were highlighted in PCoA analysis, where larval samples formed a separate cluster compared to adult samples (Permanova; *p* < 0.005; Figure 5). The Permanova analysis indicated that the bacterial communities between 1 and 15 days share similar members (Permanova; *p* > 0.05; Appendix A) compared to the other stages (1 and 5 days, 5 and 15 days) that were significantly different (Permanova; *p* < 0.05). Regarding alpha diversity, larvae showed significantly higher values in all alpha diversity indexes than the other ages. There was a significant difference between 5- and 15-day samples only in the Simpson index (Appendix A).

The *G. pallidipes* laboratory population was also dominated by Proteobacteria throughout the developmental stages (from 90.93% to 99.97%), followed by Firmicutes and Actinobacteria, the latter identified only in the larval stage (Figure 6A). Gammaproteobacteria was the most abundant class in all developmental stages, followed by Bacilli, which was found to be abundant in 5-day adults (Figure 6B). At the genus level, *Wigglesworthia* was very abundant in the adult flies (92.9 ± 5.48%, 73.4 ± 7.22% and 89.7 ± 1.17% in 1, 5, and 15 days), while in the larval stage, a high abundance of different bacterial genera was identified, including *Wiggleswothia* (61.1 ± 4.56%), *Sodalis* (12.2 ± 0.97%), *Pantoea* (9.7 ± 5.25%), and *Enterobacter* (7.7 ± 3.43%) (Figure 6C).

Potential interactions between bacterial partners were investigated with co-occurrence and mutual exclusion network analysis. The networks were visualized at the genus level and were characterized by a different number of OTUs (nodes), interactions (edges), and clustering coefficients. All three *Glossina* species were characterized by almost the same number of nodes (*G. m. morsitans* 54, *Glossina austeni* 50, and *G. pallidipes* 48). In addition, the bacterial communities of *G. m. morsitans* showed more interactions (767 edges), with a clustering coefficient of 0.307, compared to *G. austeni* (610 edges and 0.316 coefficient) and *G. pallidipes* (374 edges and coefficient of 0.270). Most of the interactions of *G. austeni* and *G. morsitans* were classified as mutual exclusions (69.38% and 66.13%, respectively), while 30.62% to 33.87% were cases of copresence. In contrast, *G. pallidipes* exhibited a different pattern (26.62% mutual exclusions and 73.38% co-presence). At the genus level, *Wigglesworthia* showed the highest degree of interaction in all three species (*G. austeni*, *G. morsitans*, and *G. pallidipes*), followed by *Sodalis* (Figure 7). A detailed list of the associations that develop between the bacterial OTUs that were identified in this study is included in the Appendix A.

### 3.3. Natural Populations Display Species- and Location-Specific Bacterial Profiles

The PCoA analysis showed that Doma samples formed a separate cluster compared to other samples, which formed a tight cluster (Permanova; *p* < 0.001; Figure 8). *Glossina morsitans* samples from the area of Doma formed a separate cluster compared to that of *G. pallidipes* from the same area (Permanova; *p* < 0.001; Figure 8). The Permanova analysis indicated also that the bacterial community in the areas of Makao and Msubugwe for *G. pallidipes* were similar (Permanova; *p* > 0.05; Appendix A). The samples from the Makao area for *G. swynnertoni* formed a separate cluster compared to the Makao samples for *G. pallidipes* (Permanova; *p* < 0.001; Figure 8). The samples of *G. pallidipes* from Doma showed significantly higher values of the Shannon index compared to the other locations. The same statistical differences were also observed for the richness and Simpson index. However, no significant differences were observed between the areas of Makao and Msubugwe. The Doma samples of *G. pallidipes* showed also significantly higher values of Shannon index and richness than Doma samples of *G. m. morsitans*. No significant differences were recorded between them according to the Simpson index. The Makao samples of *G. swynnertoni* showed also significantly higher values of Shannon index and richness than the Makao samples of *G. pallidipes*. The same statistical differences were also observed for the Simpson index (Appendix A).

Proteobacteria dominated all samples examined with a relative abundance from 88.45 to 99.96%, followed by Firmicutes (2.74% in total) and a small Actinobacteria community identified mainly in *G. swynnertoni* (Figure 9A). Gammaproteobacteria was the most abundant class, followed by Bacilli in Doma and *G. swynnertoni* from Makao (Figure 9B). At the genus level, *Wigglesworthia* was highly abundant in *G. pallidipes* (52.51 ± 7.35%, 74.46 ± 7.9%, and 89.29 ± 3.03% in Doma, Makao, and Msubugwe, respectively) and in *G. swynnertoni* (38.62 ± 8.8%). However, it was less abundant in *G. m. morsitans*, which was dominated by *Sodalis* (42.78 ± 9.69%) and *Wolbachia* (22.82 ± 5.59%)*. Acinetobacter* was mainly present in Makao, with 5.59 ± 3.79% and 11.88 ± 5.57% for *G. pallidipes* and *G. swynnertoni*, respectively (Figure 9C).

Potential interactions between bacterial partners were studied using co-occurrence and mutual exclusion network analysis, just like the laboratory samples. The networks were represented at the genus level, with varied numbers of OTUs (nodes), interactions (edges), and clustering coefficients.

For the samples from the area of Doma, *G. pallidipes* was characterized by a higher number of nodes (63 nodes) than *G. m. morsitans* (55 nodes). In addition, the bacterial communities of *G. pallidipes* showed more interactions (543 edges), with a clustering coefficient of 0.419, compared to *G. m. morsitans* (399 edges and 0.332 coefficient). Most of the interactions of *G. pallidipes* were classified as mutual exclusions (76.04%), while 23.96% were cases of co-presence, unlike *G. m. morsitans* (41.19% mutual exclusions and 58.81% co-presence). At the genus level, *Wigglesworthia* showed the highest degree of interactions in the two species (*G. m. morsitans* and *G. pallidipes*), followed by *Sodalis* (Figure 10).

For samples from the area of Makao, *G. swynnertoni* was characterized by a greater number of nodes (73 nodes) than *G. pallidipes* (51 nodes). In addition, the bacterial communities of *G. swynnertoni* showed more interactions (715 edges) with a clustering coefficient of 0.246 compared to *G. pallidipes* (325 edges and 0.238 coefficient). Most of the interactions of *G. swynnertoni* and *G. pallidipes* were classified as co-presence (51.12% and 59.04%, respectively), while 40.96% to 48.88% were cases of mutual exclusions. At the genus level, *Wigglesworthia* showed the highest degree of interactions in the two species (*G. swynnertoni* and *G. pallidipes*), followed by *Acinetobacter* and *Sodalis* (Figure 11).

For samples from the area of Msubugwe, *G. pallidipes* was characterized by 30 nodes, 152 interactions, and a clustering coefficient of 0.140. Additionally, most interactions were classified as mutual exclusions (51.56%), while 48.44% were cases of co-presence. At the genus level, *Wigglesworthia* showed the highest degree of interactions, followed by *Sodalis* (Figure 12).

### 3.4. Laboratory Populations Contain Multiple Wigglesworthia and Sodalis OTUs

For the laboratory populations, an analysis was performed for the OTUs identified as the main tsetse symbionts, *Wigglesworthia* and *Sodalis*. Five OTUs were assigned to *Wigglesworthia* (OTU1, OTU3, OTU43, OTU58, and OTU84), and two were assigned to *Sodalis* (OTU2 and OTU50). For *Wigglesworthia*, OTU1 is mostly found in larvae of all the species (43.4–59% of total *Wigglesworthia* reads in larvae) and in all three adult stages of *G. pallidipes* (95.8–99.6% of total reads in *G. pallidipes*). At the same time, OTU3 is mostly present in the adult stages of *G. m. morsitans* (94.9–97.8%), and OTU58 is mostly found in the adult stages of *G. austeni* (88.2–98.1%). OTU43 and OTU84 are found in all the species but in very small numbers. For *Sodalis,* both OTU2 and OTU50 are present in all the species, but OTU2 is more dominant than OTU50 (Appendix A).

## 4. Discussion

The results of this study on bacterial diversity, using high-throughput sequencing of the 16S rRNA gene, revealed that *Wigglesworthia* and *Sodalis* were the most abundant bacterial genera across all samples. Furthermore, some bacterial genera were more prevalent in some species than others, such as *Phyllobacterium, Pseudomonas*, and *Psychrobacter*, which were more abundant in *G. austeni* and *G. morsitans morsitans* than *G. pallidipes*. In contrast, *Bacillus* and *Lysinibacillus* were more prevalent in *G. pallidipes* than the other two species. Some bacteria were only found in one species, although in small numbers. For instance, *Acinetobacter* was found only in *G. austeni*, and *Enterococcus* was found only in *G. morsitans morsitans*. Most genera that were found were also reported in previous studies [27,28,29,30,31,33]. Overall, the three *Glossina* laboratory colonies had comparable species richness and diversity indices and comparable relative abundance across all bacterial taxa that were identified. The maternally transmitted obligate and commensal dominant symbionts, *Wigglesworthia* and *Sodalis*, as well as the environmentally acquired bacteria, exhibit comparable relative abundance values across the studied *Glossina* lab strains. This indicates that the three laboratory colonies exhibit a certain degree of homogeneity in terms of their bacterial profile, as a previous study suggested [52].

Our study established that in all analyzed samples, sex had no significant effect on the species richness/diversity of the laboratory colonies or the relative abundance of the most dominant taxa. A similar effect was observed for certain samples regarding the age factor. This is most likely due to the standard rearing conditions applied to all laboratory colonies examined, regardless of developmental stage, and can be attributed to the tsetse fly’s unique lifestyle. [27,53].

The results revealed that there were significant differences in bacterial composition between larvae and adults, suggesting that the developmental stage has an impact on the microbial diversity of tsetse flies. These differences may reflect different functional roles of symbiotic bacteria during different life stages of tsetse flies. The larval samples had significantly higher species richness and diversity than the adult gut samples. Recently, a similar significant reduction in bacterial diversity was observed in the herbivore *Spodoptera littoralis* during the development from egg to pupa, highlighting the effect the developmental stage has on the gut bacterial flora [53]. Interestingly, multiple factors such as ontogenetic stage, age, and geographic location have been shown to influence the gut microbiota in honeybees [54].

This study showed that the microbiota of wild tsetse flies was more diverse than that of laboratory populations and that the microbiota was significantly influenced by the region of origin. However, the difference observed in diversity between wild and lab tsetse strains could be related to the fact that the whole body was used for the bacterial community analysis for wild flies compared to the gut tissue that was used for the lab samples. Although a varied portion of the community resides in the gut of insects (~10^5^–10^9^ cells), bacteria can also be found in large numbers in other parts of the body, including the hemolymph, bacteriocytes, reproductive organs, etc. [55,56]. Notably, in a similar 16S amplicon sequencing analysis on *Ceratitis capitata*, De Cock et al. found no significant differences in the bacterial communities between gut-dissected samples and whole-body samples, which were preserved and surface sterilized in a similar manner [57]. Moreover, the surface sterilization method may potentially affect the studied bacterial diversity. Ethanol-treated samples could exhibit higher bacterial diversity compared to more stringent methods using bleach [58]. Bleach, on the other hand, could potentially disrupt the structure of the symbiotic bacterial communities by entering the insect’s body [59]. There are also studies that argue that surface sterilization does not alter the internal bacterial communities even when insect samples are hand collected [60]. In that case, even human bacteria, like *Staphylococcus*, were found with very low relative abundances, below 1%, which is a usual cutoff for amplicon sequencing analyses.

The most abundant bacterial genera in all samples were *Wigglesworthia*, *Acinetobacter*, and *Sodalis*, while *Wolbachia* was significantly more abundant in *G. m. morsitans* compared to the other species. *Wolbachia* is known for its ability to manipulate the reproductive biology of insects, and its presence in *G. m. morsitans* suggests a similar role in this species. Previous research indicated the well-regulated presence of beneficial partners, *Wigglesworthia* and *Sodalis*, in laboratory *G. morsitans morsitans* samples throughout host development, even after disruptive events. These events include host immune system challenges and changes in environmental conditions [61]. The higher prevalence of *Wigglesworthia* in the tsetse midgut microbiota (ranging from 85.29% to 91.19%) compared to the whole fly (ranging from 19.53% to 72.05%) is likely attributed to the dominance of *Wigglesworthia* in the midgut. This suggests that studies, that exclusively analyzed midguts, may have significantly underestimated the abundances of other bacterial taxa [62]. On the other hand, *Wolbachia* density displayed wide variability throughout development, emphasizing the importance of factors such as the age of the host during the association, the interactions with other symbionts, and the co-adaptive processes between the bacterium and the host organism [61].

*Wigglesworthia, Sodalis*, and *Wolbachia* are maternally transmitted symbionts in tsetse flies. However, *Sodalis* can also be transmitted paternally and colonize the fly during its early juvenile stages. During maternal transmission, *Wigglesworthia* and *Sodalis* are usually passed on during the larval stages via milk gland secretions and *Wolbachia* during embryogenesis via the germ cells [14,63,64]. This process occurs with high precision due to the obligate nature of the symbiotic relationship with *Wigglesworthia* [65]. In wild samples, the dominance of *Wigglesworthia* could hinder the colonization of the developing larva by other bacteria because the maternally transmitted microbes possibly occupy most of the available niches. The same could be said for the gut tissue in the laboratory strains. Paradoxically, this suggests that the tsetse fly’s immune system has evolved to accommodate these symbiotic bacteria, potentially creating opportunities for the colonization of environmental microbes capable of exploiting immune system deficiencies. Because of the unique biology of tsetse flies, environmental microbes can colonize during the pupal stages in the soil and would have to survive the process of metamorphosis in order to persist in the adult stages [27].

Regarding *Acinetobacter* and *Enterobacter*, they have been identified as having potential significance for the survival of insect hosts and the parasites they might transmit to other hosts, including humans and animals [66,67]. For example, *Acinetobacter* species are essential for the complete development of *Stomoxys calcitrans* fly larvae [67]. The Acinetobacter genus comprises potentially pathogenic opportunistic bacteria that have been previously isolated from *G. p. palpalis* in Angola and Cameroon [30,31]. In this study, they were isolated not only from *G. austeni* in laboratory populations but also from *G. pallidipes* and *G. swynnertoni* in the wild population, all savannah tsetse species. Additionally, these bacteria have been reported in the gut of *G. p. palpalis* and *G. pallicera*, as well as in *Anopheles* mosquitoes. While low populations of *Acinetobacter* species were recorded in *G. pallidipes* in Uganda, their precise role in blood-sucking insects remains unclear [68]. Furthermore, research indicates that bacteria within insect midguts can produce various antiparasitic compounds. *Enterobacter* species, for instance, can produce pigments like prodigiosin, which exhibit toxicity against *Plasmodium falciparum* and *Trypanosoma cruzi* [66,69]. Given the presence of *Enterobacter* members in the midgut of *G. p. palpalis*, further investigation into possible prodigiosin production is particularly relevant for African trypanosomiasis research. Moreover, several of the identified species are closely related to known insect pathogens, including *Pseudomonas* sp., which may have the capacity to influence the health and survival of tsetse flies [30].

*Bacillus* species are rod-shaped Gram-positive bacteria known for causing diseases like anthrax and food poisoning. Recent studies have highlighted their potential impact on the survival of insect hosts and insect vector competence [30]. These bacteria have been identified in various organisms, including the red fire ant, mosquitoes, *G. f. fuscipes*, and house flies [33,70,71,72]. These bacteria are also associated with virulence factors [73]. Another Firmicutes member, *Enterococcus*, has been found in the gut of *G. p. palpalis*, the red fire ant, and *Anopheles stephensi*. Despite these findings, the roles of *Bacillus*, *Staphylococcus*, and *Enterococcus* in tsetse flies, blood-sucking insects, remain unknown and warrant further investigation. Such research could potentially open avenues for innovative pest control strategies [68]. *Psychrobacter* species, characterized as cocci-shaped aerobic bacteria, are also present, yet their precise role in blood-sucking insects is yet to be fully elucidated [68].

The OTU co-occurrence network for each location indicated that there was high variability in the number of mutual exclusion and copresence interactions between *G. pallidipes* and the other species. Highly abundant bacterial genera, such as *Wigglesworthia* and *Sodalis*, were characterized by a high degree of interactions, suggesting their pivotal roles in the tsetse fly’s microbiota and their involvement in maintaining symbiotic relationships with other bacterial partners. Exploring the ecological implications of these associations is crucial. For example, we can identify *Wigglesworthia* and *Sodalis*, species that prefer to coexist (otu84 with otu2) or mutually exclude each other (otu58 with otu2) in hosts. Understanding these dynamics is essential for shedding light on the intricate interplay of these bacteria within the tsetse fly and how it influences the overall physiology of the host.

## 5. Conclusions

Overall, this study provides insights into the dynamics of bacterial diversity in tsetse flies and highlights the importance of considering both laboratory and wild populations in future research. Within the laboratory-reared flies, age- and species-specific variations have been identified. Lab samples demonstrated distinctive bacterial communities, which were characterized by the prevalence of certain taxa like *Wigglesworthia* and *Sodalis*. Furthermore, our investigation into natural populations has uncovered species-specific and location-specific patterns within the microbiota. These variations, particularly between wild populations, highlight the ecological and geographical factors contributing to bacterial diversity. By elucidating these patterns, our study advances our understanding of the intricate interplay between tsetse fly species, their environments, and the associated bacterial communities. The findings of this study could have important implications for the development or improvement of strategies to control tsetse fly populations, which are significant vectors of African trypanosomiasis. In this context, promising bacterial candidates can be selected and tested for probiotic effects on mass-reared flies to increase their quality. Moreover, the presence of *Wolbachia* in wild populations could be exploited for developing environmentally friendly control strategies based on reproductive parasites.

By understanding the complex interactions between the different bacterial species present in the microbiota of tsetse flies, we may be able to develop targeted interventions that disrupt the symbiotic relationship between the fly and its bacterial partners, ultimately leading to a reduction in disease transmission. Our network analyses have shed light on potential interactions among bacterial partners, emphasizing the role of dominant genera like *Wigglesworthia* and *Sodalis* in these associations. Bacterial partners like *Wolbachia*, which could potentially reduce pathogen transmission, are shown to interact with these prevalent bacterial genera. This adds a layer of complexity to our understanding of microbial interactions within tsetse flies.

## Figures and Tables

**Figure 1 insects-14-00840-f001:**
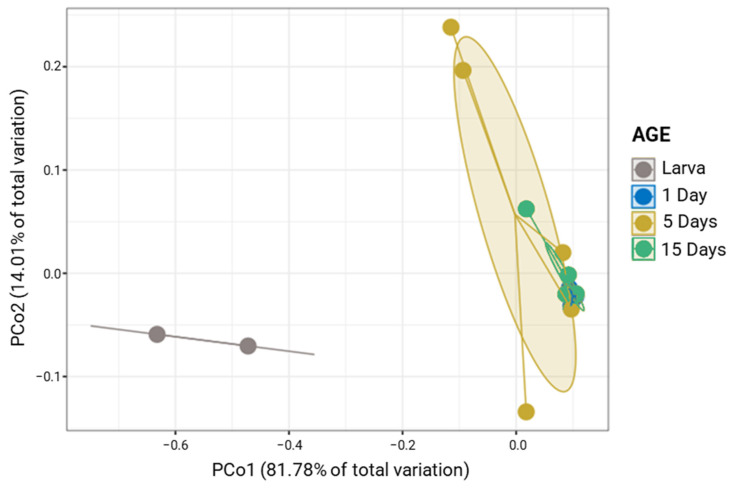
Principal coordinates analysis (PCoA) plot based on the unweighted UniFrac metric of bacterial communities of *G. austeni* samples according to the age (*p* < 0.002).

**Figure 2 insects-14-00840-f002:**
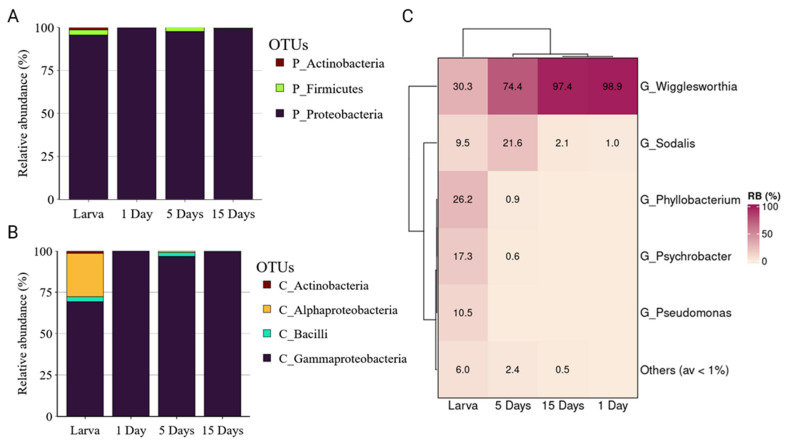
The relative abundance of bacterial communities of *Glossina austeni* at phylum (**A**), class (**B**), and genus (**C**) levels.

**Figure 3 insects-14-00840-f003:**
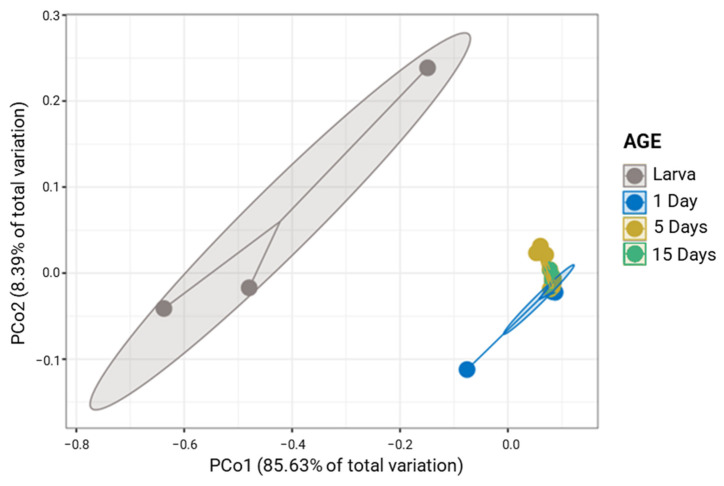
Principal coordinates analysis (PCoA) plot based on the unweighted UniFrac metric of bacterial communities of *G. morsitans morsitans* samples according to the age (*p* < 0.001).

**Figure 4 insects-14-00840-f004:**
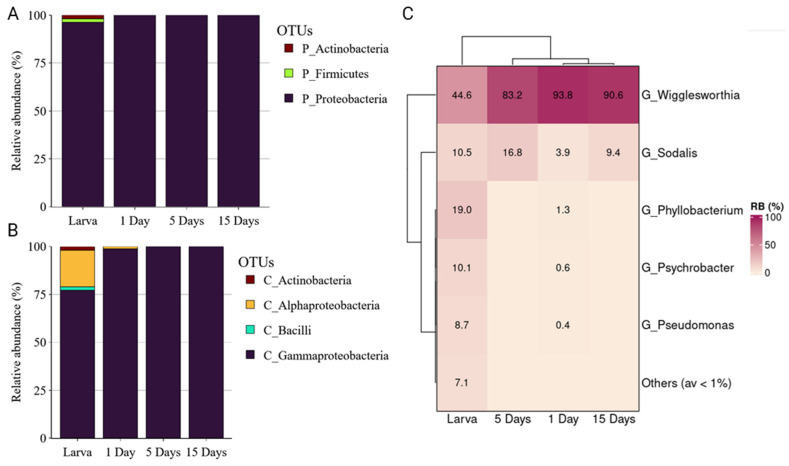
The relative abundance of bacterial communities of *Glossina morsitans morsitans* at phylum (**A**), class (**B**), and genus (**C**) levels.

**Figure 5 insects-14-00840-f005:**
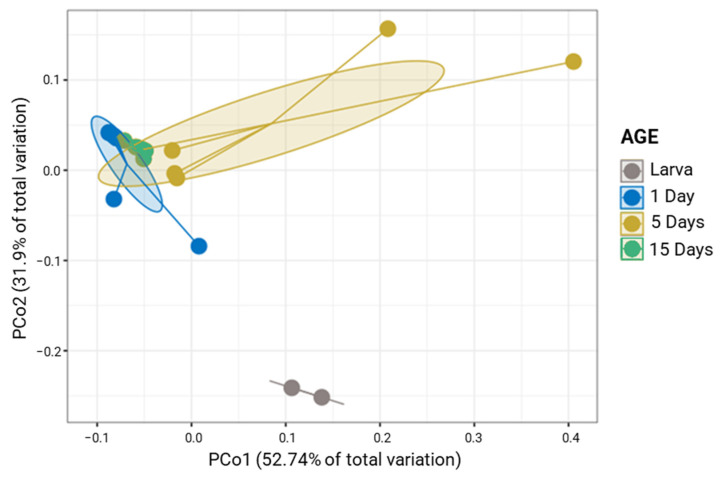
Principal coordinates analysis (PCoA) plot based on the unweighted UniFrac metric of bacterial communities for *G. pallidipes* samples according to the age (*p* < 0.007).

**Figure 6 insects-14-00840-f006:**
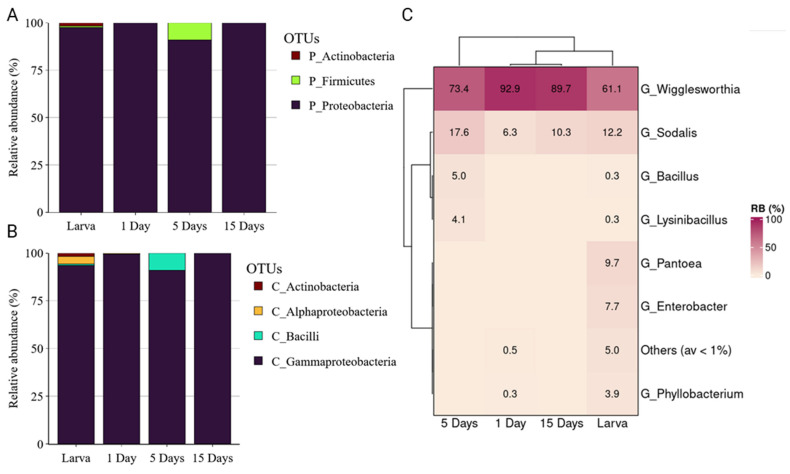
The relative abundance of bacterial communities of *G. pallidipes* at phylum (**A**), class (**B**), and genus level (**C**).

**Figure 7 insects-14-00840-f007:**
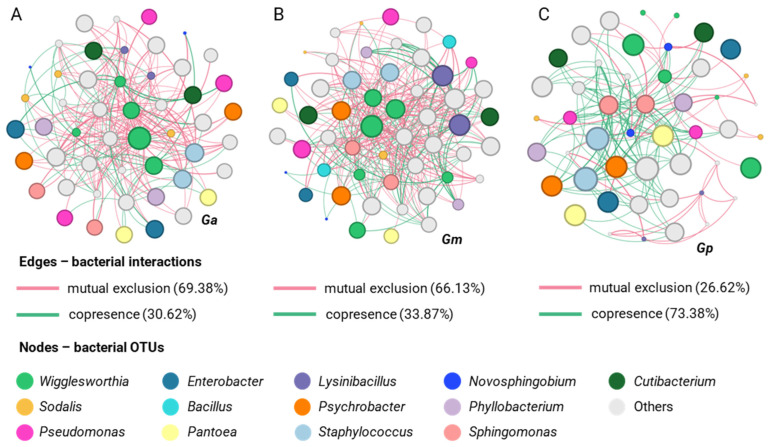
Networks displaying mutual exclusion and co-occurrence interactions between bacterial genera that make up the communities of laboratory samples of *G. austeni* (**A**), *G. m. morsitans* (**B**), and *G. pallidipes* (**C**). The size of each node is proportional to the degree of interactions. Green edges represent cases of copresence and red edges of exclusion. The numbers in parentheses describe the percentage of each type of interaction.

**Figure 8 insects-14-00840-f008:**
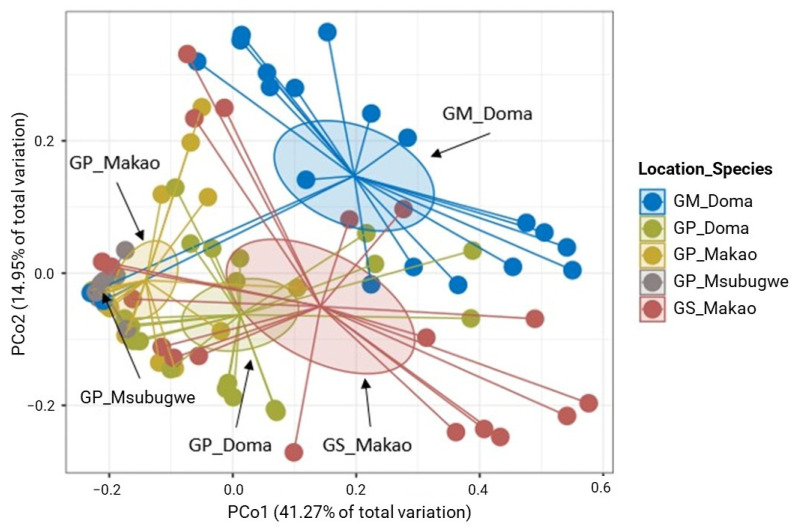
Principal coordinates analysis (PCoA) plot based on the unweighted UniFrac metric of bacterial communities for the natural samples according to species/location (*p* < 0.001).

**Figure 9 insects-14-00840-f009:**
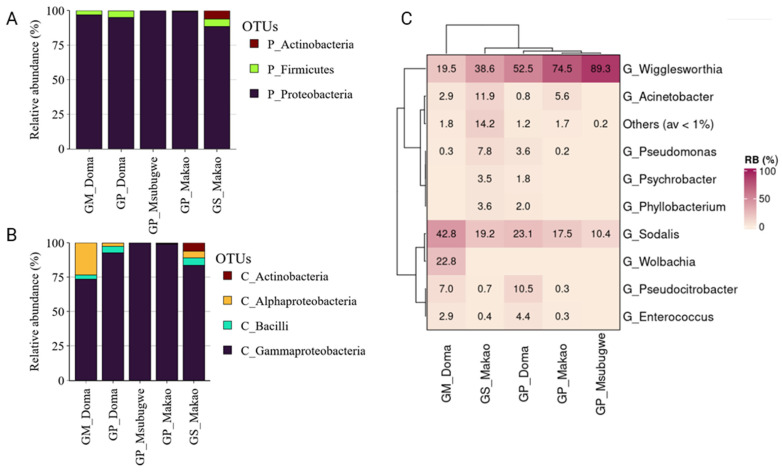
The relative abundance of bacterial communities of the wild tsetse population according to phylum (**A**), class (**B**), and genus (**C**) levels.

**Figure 10 insects-14-00840-f010:**
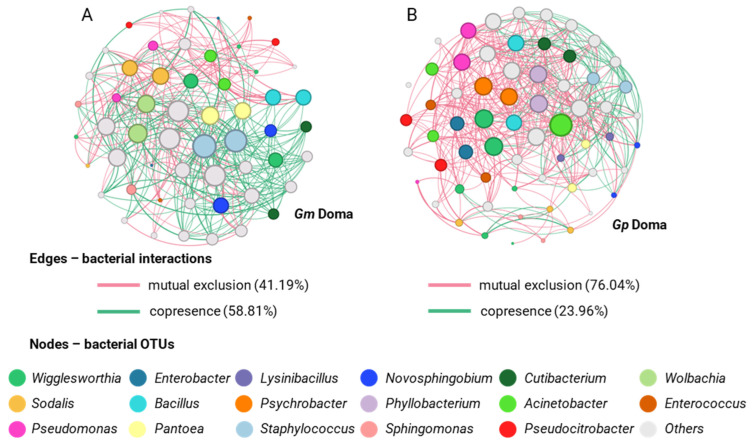
Networks displaying mutual exclusion and co-occurrence interactions between bacterial genera that compose the communities of *G. morsitans* (**A**) and *G. pallidipes* (**B**) from Doma. The size of each node is proportional to the degree of interactions. Green edges represent cases of copresence and red edges mutual exclusion. The numbers in parentheses describe the percentage of each type of interaction.

**Figure 11 insects-14-00840-f011:**
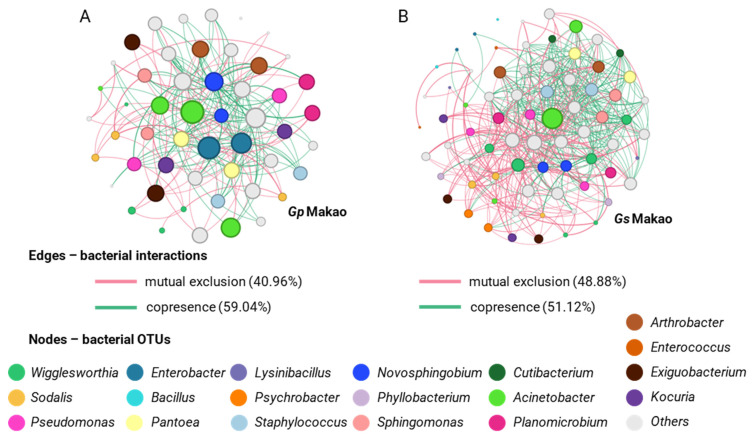
Networks displaying mutual exclusion and co-occurrence interactions between bacterial genera that compose the communities of *G. pallidipes* (**A**) and *G. swynnertoni* (**B**) from Makao. The size of each node is proportional to the degree of interactions. Green edges represent cases of copresence and red edges mutual exclusion. The numbers in parentheses describe the percentage of each type of interaction.

**Figure 12 insects-14-00840-f012:**
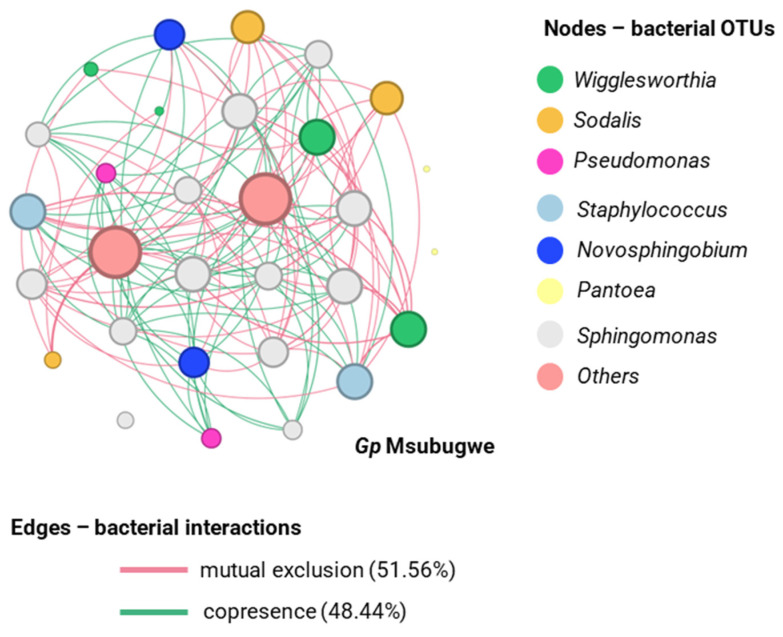
Networks displaying mutual exclusion and co-occurrence interactions between bacterial genera that compose the communities of *G. pallidipes* from Msubugwe. The size of each node is proportional to the degree of interactions. Green edges represent cases of copresence and red edges mutual exclusion. The numbers in parentheses describe the percentage of each type of interaction.

**Table 1 insects-14-00840-t001:** Sampling locations, developmental stage, and sex of studied *Glossina* samples.

Species	Location	Developmental Stage	Sex	Number of Samples
*Glossina austeni*	VVBD Insectary (Tanga, Tanzania)	Larva	-	2
		1 Day	Male	3
		5 Days		2
		15 Days		3
		1 Day	Female	3
		5 Days		3
		15 Days		3
*Glossina morsitans morsitans*	VVBD Insectary (Tanga, Tanzania)	Larva	-	3
		1 Day	Male	3
		15 Days		3
		1 Day	Female	3
		5 Days		6
		15 Days		3
	Doma (Morogoro, Tanzania)	Adult	Male	20
*Glossina pallidipes*	VVBD Insectary (Tanga, Tanzania)	Larva	-	2
		1 Day	Male	3
		5 Days		3
		15 Days		3
		1 Day	Female	3
		5 Days		3
		15 Days		3
	Doma (Morogoro, Tanzania)	Adult	Male	20
	Makao (Serengeti National Park, Tanzania)	Adult	Male	19
	Msubugwe (Pangani district)	Adult	Male	20
*Glossina swynnertoni*	Makao (Serengeti National Park, Tanzania)	Adult	Male	20

## Data Availability

All data generated from this study can be retrieved from the SRA database of NCBI under BioProject accession number PRJNA950333.

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
