# Peer review of "Characterization of the Bacterial Profile from Natural and Laboratory Glossina Populations"

_insects, 2023, doi:10.3390/insects14110840_

Round 1

Reviewer 1 Report

Comments and Suggestions for Authors

In this study, Yamlahi et al. investigated the bacterial communities in the gut of or whole body of four different Glossina spp. (tsetse flies) , well known vectors of human and animal African trypanosomiasis. Three out of four Glossina spp. were laboratory reared different developmental staged flies (larvae and adults) while remaining species was wild (male flies). Using high throughput 16S rRNA gene amplicon sequencing method, the authors demonstrated that tsetse flies harbored Wigglesworthia and Sodalis in all flies including alldevelopmental stages. Also, fly age and gender had no effect in bacterial diversity of laboratory reared each fly species. This work will be an interest of vector biologists and microbial ecologist.

Here are some major concerns about the manuscript.

1.     Method section is lacking the rationale behind choosing age (developmental stages) and sex of flies from laboratory reared flies. Also, in wild flies why authors choose only male flies? Although, both male and female flies (tsetse flies) are blood feeder, I believe that fly sex play a crucial role in bacterial communities of wild flies. If author considered both male and females would have been better to represent the bacterial communities in wild flies. Moreover, what is the reason behind dissecting gut for the lab reared flies while surface sterilization for wild flies?

2.     Analyzing data to look for the effect of fly species on bacterial communities, composition and diversity would improve the quality of the paper.

3.      In this current version, the conclusion is vague. Authors could emphasize  bacterial association specifically pathogen(s) of human and animals in each or all fly species as well as factor(s) shaping the bacterial community diversity and composition which will improve the quality.

Author Response

General comment: In this study, Yamlahi et al. investigated the bacterial communities in the gut of or whole body of four different Glossina spp. (tsetse flies), well known vectors of human and animal African trypanosomiasis. Three out of four Glossina spp. were laboratory reared different developmental staged flies (larvae and adults) while remaining species was wild (male flies). Using high throughput 16S rRNA gene amplicon sequencing method, the authors demonstrated that tsetse flies harbored Wigglesworthia and Sodalis in all flies including all developmental stages. Also, fly age and gender had no effect in bacterial diversity of laboratory reared each fly species. This work will be an interest of vector biologists and microbial ecologist.

Response: We would like to thank the reviewer for the time and effort they put into reviewing our manuscript. Their valuable feedback helped improve critical parts and the overall content of the article. We tried to incorporate the corrections suggested by the reviewer in the best way possible. Our detailed response to the reviewer’s suggestions can be found below. All suggested corrections were recorded in the revised manuscript with track changes.

Here are some major concerns about the manuscript.

Comment 1: Method section is lacking the rationale behind choosing age (developmental stages) and sex of flies from laboratory reared flies. Also, in wild flies why authors choose only male flies? Although, both male and female flies (tsetse flies) are blood feeder, I believe that fly sex play a crucial role in bacterial communities of wild flies. If author considered both male and females would have been better to represent the bacterial communities in wild flies. Moreover, what is the reason behind dissecting gut for the lab reared flies while surface sterilization for wild flies?

Response: It is common for laboratory flies to display different bacterial profiles at different developmental stages due to differences in the feed that is provided at each stage in the facility, and other biological parameters, such as the developmental process (new organs and tissues that form new niches for bacteria) and the different requirement for nutrients at each stage. On the other hand, male and female laboratory reared insects usually share similar bacterial profiles. However, we wanted to test this hypothesis also with these specific tsetse populations.

Indeed, both flies are blood feeders, and it is also quite common for bacterial profiles to be different between the two sexes in the wild. However, we focused our efforts primarily on how the bacterial profile is affected by the geographic origin. Including both sexes in the analysis would add an additional parameter that could potentially influence the composition of the bacterial profiles. Males are also of interest for mass-rearing and SIT purposes. Since sterile males have usually reduced fitness compared to wild males there is increased interest in improving the quality of mass reared males. The microbiota of insects could be a valuable tool for improving insect fitness since they can be used for nutrient provision and as probiotics in feeds. The study of the microbiota of wild males and its interactions would help us identify key players or missing elements from the lab microbiota that could help improve fitness and the SIT strategy overall.

The reason for focusing only on the gastrointestinal tissue for lab strains is similarly centered around mass rearing and the SIT technique. Identifying microbiota in the gut of lab strains could potentially reveal bacteria that are missing, or they are in fewer numbers compared to the wild strains. If they are cultivable, they could be tested for probiotic effects. Promising candidates could be used in the mass rearing process for improving fly fitness and potentially reducing the rearing cost for facilities. For wild flies we focused on the whole picture of the bacterial profile.           

We included a part in the introduction of the manuscript describing the potential use of gut bacteria and the focus on male individuals in SIT projects:

“Sterile male individuals are commonly used during this technique. Notably, the application of radiation has damaging effects on the insects, reducing their fitness and competition against wild individuals [17]. Cultivable gut bacteria with nutritional value or probiotic action, which are part of the natural flora of wild populations, could be supplemented in the artificial feed to improve the overall quality of sterile males [18,19].”

We also made modifications in the materials and methods:

Laboratory samples from different developmental stages were selected for revealing differences related to the mass rearing process (e.g., due to the different feed provided at each stage) or the biology of the insects. Such biological parameters may include the developmental process itself (e.g., creation of new niches for bacteria during the formation of tissues and organs) and the unique requirement for nutrients in each stage. The age of tsetse flies also affects infection with trypanosome [34]. For wild populations, the analysis was focused on males mainly due to their role in SIT applications. Similarly, for lab samples, the analysis was focused on the gut communities as they are of specific interest for mass breeding processes.

Comment 2: Analyzing data to look for the effect of fly species on bacterial communities, composition and diversity would improve the quality of the paper.

Response: We acknowledge the importance of considering the influence of fly species on bacterial communities’ composition, and diversity. It is common for different wild species to harbor different bacterial communities. In the manuscript, we present a comparison between species for lab samples (results, first paragraph in subsection 3.2, and in supplementary materials Table S2). As mentioned, each species harbors a different bacterial community. Table S2 also shows the composition of each community in each Glossina species. We also added a table showing the pairwise comparison results for the beta diversity analysis for the lab species (Figure S1). A similar analysis is also presented for the wild populations (results, first paragraph subsection 3.3, and supplementary materials Figures S7 and S8).

Comment 3: In this current version, the conclusion is vague. Authors could emphasize bacterial association specifically pathogen(s) of human and animals in each or all fly species as well as factor(s) shaping the bacterial community diversity and composition which will improve the quality.

Response: We have revised the conclusion and placed a stronger emphasis on bacterial associations, especially regarding the vectorial capacity of the flies, and factors influencing bacterial community diversity and composition. These aspects have been detailed to provide a more comprehensive perspective on the implications of our findings.

Reviewer 2 Report

Comments and Suggestions for Authors

Line 61 should read vector borne pathogen and not disease

Line 71 : define bilateral transmission

line 70-73: sentence too long

line 88: pathogenicity should be replaced with vectorial capacity

It might be useful to define cytoplasmic incompatibility 

Author Response

We would like to thank the reviewer for the time and effort they put into reviewing our manuscript. Their helpful comments improved the overall content of the article. We tried to incorporate the suggested corrections in the best way possible. Our detailed response to the reviewer’s suggestions can be found below. All suggested corrections were recorded in the revised manuscript with track changes.

Comment 1: Line 61 should read vector borne pathogen and not disease.

Response: The manuscript has been corrected as requested. The change has been recorded with track changes in the manuscript.

Comment 2: Line 71: define bilateral transmission.

Response: Bilateral transmission refers to the process of transmitting microorganisms or symbiotic entities, such as bacteria or viruses, from one generation to the next through both maternal and paternal means. In the context of tsetse flies and their microbiota, bilateral transmission means that these symbiotic microorganisms are passed down to the offspring not only through the mother (maternal transmission) but also through the father (paternal transmission), ensuring a high degree of inheritance and continuity of these microorganisms within the host population. To satisfy comments 2 and 3, a short description of bilateral transmission has been added to the sentence: “The tsetse microbiota is intriguing due to the flies' unique lifestyle, bilateral transmission from both parents to their progeny, involvement in reproductive strategies like cytoplasmic incompatibility, and its potential for vector and disease control.”

Comment 3: line 70-73: sentence too long

Response: The sentence has been changed to improve readability. “The tsetse microbiota is intriguing due to the flies' unique lifestyle, bilateral transmission from both parents to their progeny, involvement in reproductive strategies like cytoplasmic incompatibility, and its potential for vector and disease control.”

Comment 4: line 88: pathogenicity should be replaced with vectorial capacity.

Response: The manuscript has been corrected as requested. “The presence of the facultative symbiont Sodalis in tsetse populations is putatively related to the vectorial capacity of the fly and more specifically, its ability to transmit trypanosomes.”

Comment 5: It might be useful to define cytoplasmic incompatibility. 

Response: We agree with the reviewer’s comment and added a definition of cytoplasmic incompatibility in line 98. “Additionally, the Wolbachia strain present in G. morsitans morsitans (Gmm) can disrupt the fly’s normal reproductive activity by inducing cytoplasmic incompatibility [14], a reproductive phenomenon in some insects involving incompatibility in symbiotic microorganisms, often leading to reduced fertility or increased offspring mortality when individuals with different microbial strains mate.”

Reviewer 3 Report

Comments and Suggestions for Authors

Major revisions

-        The background section is too long, with a lot of review but with no direct correlation of a kind of research question that (is) are finally absent or do not appear clearly. The authors should clearly state why the investigate on insectary and wild populations and this will help to understand if the study design is adequate and give a meaning to the results obtained. In return, in the conclusion, the reader can’t identify a clear outcome of this study.

-        Authors didn’t explain why they only use (i) male adults from field, instead of both males and females and (ii) why only gastrointestinal tracts were used for laboratory populations while whole bodies were used in field populations. This is highly challenging when in the discussion section, lines 491 to 493, the authors say colonization by Wigglesworthia prevent other bacteria because these later occupies most of available niches; For my understanding, authors didn’t produce enough data to support this affirmation when working only on gastrointestinal tracts.

-        Moreover, all comparisons made and discussed from line 471 on the difference between microbiota of filed vs lab populations, i.e, that field tsetse microbiota was found more diverse than lab ones become obviously linked to the bias of the method. In fact, it is obvious to find a more diverse microbiome composition using the whole tsetse fly body compared to the gut, because on top of the gut, microbes may have other localizations where they may be also abundant.

-        Using ethanol may sterilize the flies before the dissection but may not remove the bacteria that are on the cuticle or prevent their DNA to be also extracted; authors should discuss on the probable presence of contaminants from environment, if they didn’t carry out any other procedure to prevent that contamination.

-        Fig 2, Fig4, Fig 6 and Fig 9: it is highly disappointing authors use default graphs without effort of reformatting the axis labels, legend fonts and ordination when necessary…. This observation is valid for almost all the graphs in the main manuscript and supplementary materials. This should absolutely corrected, as graph is not readable. Also, the general quality of the image should be ameliorated. Moreover, Actinobacteria is Phylum and Class in 2A and 2B, like in the others; I don’t know how this is possible.

-        Table1: Authors should add a column with the number of tsetse analysed for each group. This information is capital to see if they had enough flies per group to consider the results and comparisons are consistent and reliable. Looking at the Fig 3 and Fig 5 for example, it appears that the group Larvae only have 3 and 2 individuals. If this is the case, as for the other groups as well, most of the conclusions on the differences in microbiome compositions of the different groups is imprecise and may completely be different if the number of individual flies increase. Authors should therefore avoid overinterpreting their results and better discuss on these potential limitations in their study.

-        Figure 7, Fig 10, Fig 11 and 12: Although the authors took much time and pages to represented copresence and mutual exclusions between the different bacterial genera present in tsetse flies, the figures are highly confusing with the choice of colours and symbols for legends, i.e., legend symbols with squares correspond more to taxa that are circles than to interactions that are lines, also, the colours of this legend are also the same representing some taxa in the graph. Moreover, there are so many interactions (too dense network) that it would be very difficult to identify a particular couple. I will therefore recommend that the authors simplify the graph only to taxa (interactions) that may have useful interactions or to simply discard the figures from the main manuscript. Another observation is that it is obvious that the taxon Wigglesworthia that is the most abundant and the most present in samples display much interactions with others which can be present or absent. All the comments around these interactions involving Wigglesworthia are useless unless the authors prove that some may be interesting even from literature.

-        Manuscript should be read and edited to improve the language.

Minor revisions

-        Why only wild males? Please explain in methods, sample overview why wild females were not used.

-        Line 159: please change “a modified version of…” to “a modified protocol using….”

-       -        Line 163-164: I couldn’t identify the primers U341F-MiSeq and U805R-MiSeq in the reference (24: doi:10.1093/nar/gks808) provided by the authors. Please check and confirm or specify the right one!!!

-        Line 166: is it 0.2 µL of each dNTP (25 mM) or of a mixed dNTPs Solution? Please make it clear

-        Line 176-177: Reword “in a manner similar to 176 that employed…” to “as proceeded…”

-        Line 183 Specify who is Macrogen

-        Line 184 after Miseq platform, specity (Company, Town, and country) and specify each or mixed dNTPs solution as previously in line 166

-        Line 192: rephrase the sentence, putting “According to the manufacturer's instructions” at the end.

-        Line 194 give reference to the company of the selection kit and do the same for all kits or equipment or…

-        Line 195 … to “determine” the concentration, not detect

-        Line 210 211: Please specify the threshold used to discard sequences with low abundance

-        Data analyses: please give references to all packages used for the analyses.

-        Line 225 226: Rephrase the sentence, starting with “Co-occurrence…..” and add space between The and CoNet

-        Line 225 226: Change “The building of the network was based….” To “The network was built based…”

-        Line 225: Remove The before “Statistical….

-        Line 252: Replace “we discovered that certain OTUs… by “we found that…

-        253: delete often

-        257 to 259: please provide a statistic for the comparisons; also change “although in small numbers” to “although in small relative abundance”.

-        266: remove The same before Statistical…..

-        Page 7: Please reformat the Fig1, using distinctive colours i.e. for 5 days and 15 days adults and ordination in the legend, i.e., larva- 1days-5days-15days. This help better reading and understanding

-        272-274: it is confusing to describe the main phyla for G. austeni as a whole but secondary phyla only for larva. You can split and give results for adults and larvae separately.

-        287 and 288: remove “s” on Figures S3

-        290: “The same statistical differences were also observed for all other indexes” – Authors should be specific on the indices they are talking about and state the biological meaning of that difference. Also, a new sentence cannot start with “But”. Moreover, that sentence is poorly written regarding the language or syntax

-        296: Start the sentence by “Glossina morsitans morsitans laboratory….”

-        299 and 301: Replace “very abundant” by “highly abundant”

-        308: start with “Glossina pallidipes….

-        Figure 4: please, delete “D_digit_” on taxa names, I am not sure it brings other information than confusion on the graph

-        313 314, idem as comment for line 290

-        Fig 6: Same comment as for Fig 4 and general comments on Figs

-        332: What are bacterial partners??

-        336: For me, 54, 50 and 48 are not different unless there is a statistic to prove it

-        351, title 3.3 modify to: Natural populations display a species- and location- specific bacterial profiles  

-        Line 427: correct Wigglesworthia. Authors cannot affirm here that different OTUs identified represent different species. I would recommend not to over interpret results.

-        Line 428: You mean population”s”??? or which one?

-        Line 435: add space between G. and austeni

-        Line 448 449: “…same species richness and diversity indices…” Authors could plot Shannon or Chao 1 or other indices and compare in the results section to show how these are the same for all colonies. I am afraid I didn’t see these results.

-        Line 452: Please be specific on what you mean by “comparable dynamics”.

-        483 484: “Previous studies have shown that the bacterial community of tsetse flies is characterised by the presence of Wigglesworthia, Sodalis and Wolbachia” Can authors give references of these studies, since of my knowledge, many studies also failed to identify the two later or did, but with very small abundance. Authors should also discuss on the variability of the results obtained elsewhere.

-        509 510: “The findings of this study have important implications for the development of strategies to control tsetse fly populations, which are significant vectors of African trypanosomiasis”, please be clear or specific providing concrete example of benefits of this work

Comments on the Quality of English Language

The paper should be read and edited to improve the language. Some examples are highlighted in minor revisions.

Author Response

We appreciate the opportunity to revise and improve our manuscript and would like to thank the reviewer for the time and effort they put into the reviewing process. We tried to incorporate the suggested corrections in the best way possible. An extensive point-by-point response to the reviewer’s comments can be found below. All suggested corrections were recorded in the revised manuscript with track changes. We believe that the revisions implemented based on the reviewer’s comments have strengthened the manuscript.

Comment 1: The background section is too long, with a lot of review but with no direct correlation of a kind of research question that (is) are finally absent or do not appear clearly. The authors should clearly state why the investigate on insectary and wild populations and this will help to understand if the study design is adequate and give a meaning to the results obtained. In return, in the conclusion, the reader can’t identify a clear outcome of this study.

Response: The descriptive part of the introduction was reduced significantly. However, we added some parts that were requested by other reviewers. Overall, the introduction is more compact. The focus of the current study is the analysis of the symbiotic bacterial profiles of wild and laboratory populations of different tsetse species, as well as the analysis of associations between the members that constitute each community. We also made changes to the aim of the study to justify why wild and lab strains were analyzed. We also revised the conclusion according to the reviewer’s comment to highlight these reasons and make the conclusion less vague.

Comment 2: Authors didn’t explain why they only use (i) male adults from field, instead of both males and females and (ii) why only gastrointestinal tracts were used for laboratory populations while whole bodies were used in field populations. This is highly challenging when in the discussion section, lines 491 to 493, the authors say colonization by Wigglesworthia prevent other bacteria because these later occupies most of available niches; For my understanding, authors didn’t produce enough data to support this affirmation when working only on gastrointestinal tracts.

Response: Male individuals are the main interest for mass-rearing and SIT purposes. Since sterile males have usually reduced fitness compared to wild males there is increased interest in improving the quality of mass reared males. The microbiota of insects could be a valuable tool for improving insect fitness since they can be used for nutrient provision and as probiotics in feeds. The study of the microbiota of wild males and its interactions would help us identify key players or missing elements from the lab microbiota that could help improve fitness and the SIT strategy overall. Additionally, we focused our efforts primarily on how the bacterial profile is affected by the geographic origin. Including both sexes in the analysis would add an additional parameter that could potentially influence the composition of the bacterial profiles.

The reason for focusing only on the gastrointestinal tissue for lab strains is similarly centered around mass rearing and the SIT technique. Identifying microbiota in the gut of lab strains could potentially reveal bacteria that are missing, or they are in fewer numbers compared to the wild strains. If they are cultivable, they could be tested for probiotic effects. Promising candidates could be used in the mass rearing process for improving fly fitness and potentially reducing the rearing cost for facilities. For wild flies we focused on the whole picture of the bacterial profile.

We included a part in the introduction of the manuscript describing the potential use of gut bacteria and the focus on male individuals in SIT projects:

“Sterile male individuals are commonly used during this technique. Notably, the application of radiation has damaging effects on the insects, reducing their fitness and competition against wild individuals [17]. Cultivable gut bacteria with nutritional value or probiotic action, which are part of the natural flora of wild populations, could be supplemented in the artificial feed to improve the overall quality of sterile males [18,19].”

We also made modifications in the materials and methods:

Laboratory samples from different developmental stages were selected for revealing differences related to the mass rearing process (e.g., due to the different feed provided at each stage) or the biology of the insects. Such biological parameters may include the developmental process itself (e.g., creation of new niches for bacteria during the formation of tissues and organs) and the unique requirement for nutrients in each stage. The age of tsetse flies also affects infection with trypanosome [34]. For wild populations, the analysis was focused on males mainly due to their role in SIT applications. Similarly, for lab samples, the analysis was focused on the gut communities as they are of specific interest for mass breeding programs.

We also made modifications in the discussion:

In wild samples, the dominance of Wigglesworthia could hinder the colonization of the developing larva by other bacteria, because the maternally transmitted microbes possibly occupy most of the available niches. The same could be said for the gut tissue in the laboratory strains.

Comment 3: Moreover, all comparisons made and discussed from line 471 on the difference between microbiota of filed vs lab populations, i.e, that field tsetse microbiota was found more diverse than lab ones become obviously linked to the bias of the method. In fact, it is obvious to find a more diverse microbiome composition using the whole tsetse fly body compared to the gut, because on top of the gut, microbes may have other localizations where they may be also abundant.

Response: We modified the discussion to consider this bias in the comparison of bacterial communities between wild and lab samples. “However, the difference observed in diversity between wild and lab tsetse strains could be related to the fact that the whole body was used for the bacterial community analysis for wild flies compared to the gut tissue that was used for the lab samples. Although a varied portion of the community resides in the gut of insects (~105-109 cells), bacteria can also be found in large numbers in other parts of the body, including the haemocoel, bacteriocytes, reproductive organs etc. [66,67]. Notably, in a similar 16S amplicon sequencing analysis on Ceratitis capitata, De Cock et al. found no significant differences in the bacterial communities between gut dissected samples and whole body samples, which were preserved and surface sterilized in a similar manner [68].”.

“Previous research indicated the well-regulated presence of beneficial partners, Wigglesworthia and Sodalis, in laboratory G. morsitans morsitans samples throughout host development, even after disruptive events. These events include host immune system challenges and changes in environmental conditions [52]. The higher prevalence of Wigglesworthia in the tsetse midgut microbiota (ranging from 85.29% to 91.19%) compared to the whole fly (ranging from 19.53% to 72.05%) is likely attributed to the dominance of Wigglesworthia in the midgut. This suggests that studies, which exclusively analysed midguts, may have significantly underestimated the abundances of other bacterial taxa [53].”.

Comment 4: Using ethanol may sterilize the flies before the dissection but may not remove the bacteria that are on the cuticle or prevent their DNA to be also extracted; authors should discuss on the probable presence of contaminants from environment, if they didn’t carry out any other procedure to prevent that contamination.

Response: Indeed, surface sterilization with ethanol cannot remove all the bacterial DNA from the cuticle. Traces of DNA will remain and may affect diversity analyses. At the same time, more robust methods, like laboratory bleach, have been shown to disrupt the endosymbiotic bacterial communities. We added a passage in the discussion regarding this issue: “Moreover, the surface sterilization method may potentially affect the studied bacterial diversity. Ethanol treated samples could exhibit higher bacterial diversity compared to more stringent methods using bleach [66]. Bleach, on the other hand, could potentially disrupt the structure of the symbiotic bacterial communities by entering the insect’s body [67]. There are also studies that argue that surface sterilization does not alter the internal bacterial communities even when insect samples are hand collected [68]. In that case, even human bacteria, like Staphylococcus, were found with very low relative abundances, below 1% which is a usual cutoff for amplicon sequencing analyses.”

Comment 5: Fig 2, Fig4, Fig 6 and Fig 9: it is highly disappointing authors use default graphs without effort of reformatting the axis labels, legend fonts and ordination when necessary…. This observation is valid for almost all the graphs in the main manuscript and supplementary materials. This should absolutely corrected, as graph is not readable. Also, the general quality of the image should be ameliorated. Moreover, Actinobacteria is Phylum and Class in 2A and 2B, like in the others; I don’t know how this is possible.

Response: We improved quality and general appearance for most figures and plots in the manuscript. We tried to increase resolution, size, repaired axes, labels, and legends. Regarding the Phylum Actinobacteria, the version of Silva database that was used for the analysis separates them into 6 classes Actinobacteria, Acidimicrobiia, Coriobacteriia, Nitriliruptoria, Rubrobacteria, and Thermoleophilia. Figures now contain the letter P_Actinobacteria to refer to the Phylum and C_Actinobacteria to refer to the Class. Other databases, the 16S database of NCBI, or the newest Silva database may have updated/changed the taxonomic classification of Actinobacteria.

Comment 6: Table1: Authors should add a column with the number of tsetse analysed for each group. This information is capital to see if they had enough flies per group to consider the results and comparisons are consistent and reliable. Looking at the Fig 3 and Fig 5 for example, it appears that the group Larvae only have 3 and 2 individuals. If this is the case, as for the other groups as well, most of the conclusions on the differences in microbiome compositions of the different groups is imprecise and may completely be different if the number of individual flies increase. Authors should therefore avoid overinterpreting their results and better discuss on these potential limitations in their study.

Response: We updated the Table to contain information about the number of samples per species. We also added information about the samples in the materials and methods: “Each lab sample was a pool of five tissues, while wild samples consisted of individual insects.”.

Comment 7:  Figure 7, Fig 10, Fig 11 and 12: Although the authors took much time and pages to represented copresence and mutual exclusions between the different bacterial genera present in tsetse flies, the figures are highly confusing with the choice of colours and symbols for legends, i.e., legend symbols with squares correspond more to taxa that are circles than to interactions that are lines, also, the colours of this legend are also the same representing some taxa in the graph. Moreover, there are so many interactions (too dense network) that it would be very difficult to identify a particular couple. I will therefore recommend that the authors simplify the graph only to taxa (interactions) that may have useful interactions or to simply discard the figures from the main manuscript. Another observation is that it is obvious that the taxon Wigglesworthia that is the most abundant and the most present in samples display much interactions with others which can be present or absent. All the comments around these interactions involving Wigglesworthia are useless unless the authors prove that some may be interesting even from literature.

Response: We updated and improved the resolution, legends, and overall quality of network images. We also added a table in the supplementary materials that lists all the interactions between bacterial OTUs in all samples at the species level. The table is very useful for anyone willing to check for specific interactions within each sample. It is true that network images are too dense and are better viewed in the interactive environment of appropriate software. However, we believe that they provide a general idea about the organization of the bacterial communities, their complexity and structure, and we decided to keep them in the main manuscript. We also restructured the passage about the network interactions in the discussion.   

Comment 8: Manuscript should be read and edited to improve the language.

Response: We tried to improve readability and the language in many parts of the manuscript.

Minor revisions

Comment 9: Why only wild males? Please explain in methods, sample overview why wild females were not used.

Response: We included a part in the introduction of the manuscript describing the potential use of gut bacteria and the focus on male individuals in SIT projects:

“Sterile male individuals are commonly used during this technique. Notably, the application of radiation has damaging effects on the insects, reducing their fitness and competition against wild individuals [17]. Cultivable gut bacteria with nutritional value or probiotic action, which are part of the natural flora of wild populations, could be supplemented in the artificial feed to improve the overall quality of sterile males [18,19].”

We also made modifications in the materials and methods:

For wild populations, the analysis was focused on males mainly due to their role in SIT applications.

Comment 10:  Line 159: please change “a modified version of…” to “a modified protocol using….”

Response: The manuscript has been corrected as requested.

Comment 11:  Line 163-164: I couldn’t identify the primers U341F-MiSeq and U805R-MiSeq in the reference (24: doi:10.1093/nar/gks808) provided by the authors. Please check and confirm or specify the right one!!!

Response: The primers exist in the extensive supplementary materials of that study. Since it is a tedious task to search for them manually, we added their sequence in the manuscript.

5’-TCGTCGGCAGCGTCAGATGTGTATAAGAGACAG(CCTACGGGRSGCAGCAG)-3′ and U805R-MiSeq 5’-GTCTCGTGGGCTCGGAGATGTGTATAAGAGACA(GGACTACHVGGGTATCTAATCC)-3′

(Parentheses indicate the primer region)

Comment 12: Line 166: is it 0.2 µL of each dNTP (25 mM) or of a mixed dNTPs Solution? Please make it clear

Response: It is a mixed dNTPs solution. We added “a mixed solution” to the article.

Comment 13: Line 176-177: Reword “in a manner similar to 176 that employed…” to “as proceeded…”

Response: The manuscript has been corrected as requested.

Comment 14:  Line 183 Specify who is Macrogen

Response: The manuscript has been corrected as requested. We added “Inc. (Seoul, South Korea)”

Comment 15:  Line 184 after Miseq platform, specity (Company, Town, and country) and specify each or mixed dNTPs solution as previously in line 166

Response: The manuscript has been revised as requested.

Comment 16:  Line 192: rephrase the sentence, putting “According to the manufacturer's instructions” at the end.

Response: The sentence was rephrased.

Comment 17:  Line 194 give reference to the company of the selection kit and do the same for all kits or equipment or…

Response: A reference was added

Comment 18:  Line 195 … to “determine” the concentration, not detect.

Response: The manuscript has been corrected as requested.

Comment 19:  Line 210 211: Please specify the threshold used to discard sequences with low abundance.

Response: The threshold is included in the first paragraph of the results. “…were identified with a relative abundance greater than 0.1%”

Comment 20:  Data analyses: please give references to all packages used for the analyses.

Response: We added missing references for vegan, ggplot2 and FastTree.

Comment 21:  Line 225 226: Rephrase the sentence, starting with “Co-occurrence…..” and add space between The and CoNet

Response: The sentence was rephrased: Co-occurrence networks of OTUs were generated using CoNet, a plugin for Cytoscape 3.8.2.

Comment 22:  Line 225 226: Change “The building of the network was based….” To “The network was built based…”

Response: The manuscript has been corrected as requested.

Comment 23:  Line 225: Remove The before “Statistical….

Response: The manuscript has been corrected as requested.

Comment 24:  Line 252: Replace “we discovered that certain OTUs… by “we found that…

Response: The manuscript has been corrected as requested.

Comment 25: 253: delete often.

Response: The manuscript has been corrected as requested.

Comment 26:  257 to 259: please provide a statistic for the comparisons; also change “although in small numbers” to “although in small relative abundance”.

Response: Added statistics and revised the sentence.

Comment 27: 266: remove The same before Statistical…..

Response: The manuscript has been corrected as requested.

Comment 28: Page 7: Please reformat the Fig1, using distinctive colours i.e. for 5 days and 15 days adults and ordination in the legend, i.e., larva- 1days-5days-15days. This help better reading and understanding

Response: The figure was fixed

Comment 29: 272-274: it is confusing to describe the main phyla for G. austeni as a whole but secondary phyla only for larva. You can split and give results for adults and larvae separately.

Response: For G. austeni, larvae contain the most diverse community. The other 3 ages are dominated mostly by gammaproteobacteria.

Comment 30: 287 and 288: remove “s” on Figures S3

Response: The manuscript has been corrected as requested.

Comment 31: 290: “The same statistical differences were also observed for all other indexes” – Authors should be specific on the indices they are talking about and state the biological meaning of that difference. Also, a new sentence cannot start with “But”. Moreover, that sentence is poorly written regarding the language or syntax.

Response: The text was revised.

Comment 32: 296: Start the sentence by “Glossina morsitans morsitans laboratory….”

Response: The text was revised.

Comment 33: 299 and 301: Replace “very abundant” by “highly abundant”.

Response: The manuscript has been corrected as requested.

Comment 34: 308: start with “Glossina pallidipes….

Response: The text was revised accordingly.

Comment 35: Figure 4: please, delete “D_digit_” on taxa names, I am not sure it brings other information than confusion on the graph.

Response: We revised the figure. “D_digit_” were changed to P for phylum, C for class, and G for genus.

Comment 36: 313 314, idem as comment for line 290

Response: We revised the text as requested.

Comment 37: Fig 6: Same comment as for Fig 4 and general comments on Figs

Response: We revised both figures 6 and 5. Probably the comment was referring to figure 5.

Comment 38: 332: What are bacterial partners??

Response: It refers to bacterial taxa forming different types of associations. We wrote a short description at the first occurrence of the word in the introduction.

Comment 39: 336: For me, 54, 50 and 48 are not different unless there is a statistic to prove it

Response: Indeed, they are the same order of magnitude. The sentence was revised.

Comment 40: 351, title 3.3 modify to: Natural populations display a species- and location- specific bacterial profiles.  

Response: The manuscript has been corrected as requested.

Comment 41: Line 427: correct Wigglesworthia. Authors cannot affirm here that different OTUs identified represent different species. I would recommend not to over interpret results.

Response: The Wigglesworthia in the title was corrected. The similarity cutoff for clustering OTUs as mentioned in the analysis was set at 97%. These OTUs have less than 97% similarity in the fragment of the 16S that was sequenced (v3-v4), which translates into more than 15 different bases in the sequence. Such differences could potentially translate into more than 50 different bp in the complete 16S molecule. Most probably these OTUs constitute different species. However, since it is not certain we changed the title to OTUs instead of species.

Comment 42: Line 428: You mean population”s”??? or which one?

Response: We mean populations. It has been corrected.

Comment 43: Line 435: add space between G. and austeni

Response: The manuscript has been corrected as requested.

Comment 44: Line 448 449: “…same species richness and diversity indices…” Authors could plot Shannon or Chao 1 or other indices and compare in the results section to show how these are the same for all colonies. I am afraid I didn’t see these results.

Response: Alpha diversity plots are included in the supplementary materials.

Comment 45: Line 452: Please be specific on what you mean by “comparable dynamics”.

Response: Revised to “…exhibit comparable relative abundance values across the studied Glossina lab strains”.

Comment 46: 483 484: “Previous studies have shown that the bacterial community of tsetse flies is characterised by the presence of Wigglesworthia, Sodalis and Wolbachia” Can authors give references of these studies, since of my knowledge, many studies also failed to identify the two later or did, but with very small abundance. Authors should also discuss on the variability of the results obtained elsewhere.

Response: The discussion was revised as requested.

Comment 47: 509 510: “The findings of this study have important implications for the development of strategies to control tsetse fly populations, which are significant vectors of African trypanosomiasis”, please be clear or specific providing concrete example of benefits of this work.

Response: The conclusions were revised as requested.

Reviewer 4 Report

Comments and Suggestions for Authors

The manuscript by Yamlahi et al. describes the sequencing and analysis of the microbial composition of tsetse flies from wild and lab populations. The study expands on an already significant body of knowledge on the microbiome of tsetse flies and adds by making comparisons between different developmental stages and wild versus lab populations. Overall the paper is interesting, but has some issues that need to be addressed prior to publication. I have listed these issues below.

Major Comments:

Introduction: I recommend revising the introduction to include a description of the development of the drug fexidinazol. This would provide a more comprehensive background for readers.

Additionally, it would be advisable to remove references to insecticide resistance in tsetse flies, as this has not been observed in wild populations and could mislead the reader.

Materials and Methods: The section lacks clarity on the number of replicates per sample type. This is crucial for assessing the robustness of the experimental design and the results. Please revise to include this information.

Raw sequencing data should be made publicly available for transparency and further research. Please specify whether the data have been deposited in a data repository and include reference information in the paper.

Results: There is ambiguity regarding the 5 different OTUs associated with Wigglesworthia. It's essential to clarify whether these represent different strains of Wigglesworthia or are indicative of random point mutations within Wigglesworthia populations.

Concerning DNA extractions, using ethanol for surface sterilization will kill external microbes but will not effectively remove microbial DNA. This could potentially impact your results. Revising the methodology and potentially the results to account for this is recommended.

Figures: Please revise the figures to increase the font size of the labels. As they stand, they are difficult to read and detract from the overall impact of your findings.

Discussion: While you have listed the bacteria identified, a deeper discussion of the functional and ecological implications of these bacteria would provide more depth to the paper. Specifically, elaborating on the functional relevance of mutually exclusive or co-present bacteria in different groups could add nuance to your discussion.

Consider the analytical possibility of subtracting Wigglesworthia from your analyses. Given its ubiquity in tsetse flies and dominant presence in the samples, removing it could allow for the identification of more subtle differences between lab colonies and field sites.

 Minor Comments: In the simple summary on line 4, the phrase "inhabit the internal of the flies’' body" should be corrected to "inhabit the internal organs of the flies' body" for anatomical accuracy.

Summary: The paper contributes valuable data but requires significant revisions for clarity, depth, and methodological robustness. Addressing these points will greatly enhance the paper's clarity and scientific contribution.

Comments on the Quality of English Language

The paper should be proofed to catch some minor grammatical issues.

Author Response

General comment: The manuscript by Yamlahi et al. describes the sequencing and analysis of the microbial composition of tsetse flies from wild and lab populations. The study expands on an already significant body of knowledge on the microbiome of tsetse flies and adds by making comparisons between different developmental stages and wild versus lab populations. Overall, the paper is interesting, but has some issues that need to be addressed prior to publication. I have listed these issues below.

Response: We would like to thank the reviewer for the time and effort they put into reviewing our manuscript. Their helpful feedback improved the overall content of the article. We tried to incorporate the suggested corrections in the best way possible. Our detailed response to the reviewer’s suggestions can be found below. All suggested corrections were recorded in the revised manuscript with track changes.

Major Comments:

Introduction

Comment 1: I recommend revising the introduction to include a description of the development of the drug fexidinazol. This would provide a more comprehensive background for readers.

Response: Thank you for your recommendation. We added a paragraph about the drug fexidinazol in line 68.

Comment 2: Additionally, it would be advisable to remove references to insecticide resistance in tsetse flies, as this has not been observed in wild populations and could mislead the reader.

Response: The manuscript has been corrected as requested.

Materials and Methods

Comment 3: The section lacks clarity on the number of replicates per sample type. This is crucial for assessing the robustness of the experimental design and the results. Please revise to include this information.

Response: we added the number of samples to the table.

Comment 4: Raw sequencing data should be made publicly available for transparency and further research. Please specify whether the data have been deposited in a data repository and include reference information in the paper.

Results: All data generated from this study can be retrieved from the SRA database of NCBI under BioProject accession number PRJNA950333. This information was already mentioned at the end of the manuscript, in the data availability statement. However, it was also added to the materials and methods section.

Comment 5: There is ambiguity regarding the 5 different OTUs associated with Wigglesworthia. It's essential to clarify whether these represent different strains of Wigglesworthia or are indicative of random point mutations within Wigglesworthia populations.

Response: The similarity cutoff for clustering OTUs as mentioned in the analysis was set at 97%. These OTUs have less than 97% similarity in the fragment of the 16S that was sequenced (v3-v4), which translates into more than 15 different bases in the sequence. Such differences could potentially translate into more than 50 different bp in the complete 16S molecule. Most probably these OTUs constitute different species. However, since it is not certain we changed the title to OTUs instead of species.

Comment 6: Concerning DNA extractions, using ethanol for surface sterilization will kill external microbes but will not effectively remove microbial DNA. This could potentially impact your results. Revising the methodology and potentially the results to account for this is recommended.

Response: Indeed, this is true for surface sterilization with ethanol. We have included a passage in the discussion regarding surface sterilization of the samples and the implications on the results presented in the manuscript.

Comment 7: Figures: Please revise the figures to increase the font size of the labels. As they stand, they are difficult to read and detract from the overall impact of your findings.

Response: We revised all figures in the manuscript and supplementary materials. We increased size, resolution, changed ambiguous colors, improved legends, and axes.

Discussion

Comment 8: While you have listed the bacteria identified, a deeper discussion of the functional and ecological implications of these bacteria would provide more depth to the paper. Specifically, elaborating on the functional relevance of mutually exclusive or co-present bacteria in different groups could add nuance to your discussion.

Response: the discussion was revised according to the reviewer’s comment.

Comment 9: Consider the analytical possibility of subtracting Wigglesworthia from your analyses. Given its ubiquity in tsetse flies and dominant presence in the samples, removing it could allow for the identification of more subtle differences between lab colonies and field sites.

Response: Despite the dominance of Wigglesworthia and its high relative abundance in almost all samples, the remaining important factors of the bacterial community are not masked. This is because the lower cutoff was set at 0.1% rather than the typical 1% cutoff that is used in most 16S amplicon sequencing community analyses.

Minor Comments:

Comment 10: In the simple summary on line 4, the phrase "inhabit the internal of the flies’' body" should be corrected to "inhabit the internal organs of the flies' body" for anatomical accuracy.

Response: The manuscript has been corrected as requested.

Summary: The paper contributes valuable data but requires significant revisions for clarity, depth, and methodological robustness. Addressing these points will greatly enhance the paper's clarity and scientific contribution.

Round 2

Reviewer 1 Report

Comments and Suggestions for Authors

All issues raised previously are addressed.

Reviewer 4 Report

Comments and Suggestions for Authors

Thank you for the revised version of your manuscript. It is much improved and appropriate for publication.